# Transdermal Delivery of Metformin Using Dissolving Microneedles and Iontophoresis Patches for Browning Subcutaneous Adipose Tissue

**DOI:** 10.3390/pharmaceutics14040879

**Published:** 2022-04-17

**Authors:** Mehrnaz Abbasi, Zhaoyang Fan, John A. Dawson, Shu Wang

**Affiliations:** 1Department of Nutritional Sciences, Texas Tech University, Lubbock, TX 79409, USA; mehrnaz.abbasi@asu.edu (M.A.); john.a.dawson@ttu.edu (J.A.D.); 2College of Health Solutions, Arizona State University, Phoenix, AZ 85004, USA; 3School of Electrical, Computer and Energy Engineering, Arizona State University, Tempe, AZ 85281, USA; zhaoyang.fan@asu.edu

**Keywords:** obesity, transdermal delivery, microneedle, iontophoresis, adipose tissue browning, metformin

## Abstract

Obesity is a serious public health problem that is strongly associated with increased multiple comorbidities such as diabetes, cardiovascular disease, and some types of cancer. While current anti-obesity treatments have various issues, locally transforming energy-storing white adipose tissue (WAT) into energy-burning brown-like/beige adipose tissue, the so-called browning of WAT, has been suggested to enhance obesity treatment efficiency with minimized side effects. Metformin is a first-line antidiabetes drug and a potent activator of AMP-activated protein kinase. Emerging evidence has suggested that metformin might enhance energy expenditure via the browning of WAT and hence reduce body weight. Subcutaneous WAT is easier to access and has a stronger browning potential than other WAT depots. In this study, we used dissolvable poly (lactic-co-glycolic acid) microneedles (MN) to deliver metformin to the subcutaneous WAT in obese C57BL/6J mice with the assistance of iontophoresis (INT), and then investigated metformin-induced WAT browning and its subsequent thermogenesis effects. Compared with MN alone or INT alone, MN + INT had better anti-obesity activity, as indicated by decreasing body weight and fat gain, increased energy expenditure, decreased fat pad size, and improved energy metabolism through the browning of WAT. Browning subcutaneous WAT by delivering metformin and other browning agents using this MN + INT approach might combat obesity in an effective, easy, and safe regimen.

## 1. Introduction

Obesity, excess body fat accumulation, is one of the most challenging public health problems and is strongly associated with various comorbidities such as Type 2 diabetes, cardiovascular disease, and some types of cancer [1,2]. Healthy diets, exercise, and lifestyle interventions are commonly recommended for fighting obesity. However, they are hard to follow and have low efficacy in the long term. The existing anti-obesity medical therapies, including pharmacotherapy and surgery, have some limitations. Orally administered medications that target energy intake by reducing appetite or lowering nutrition absorption are commonly used in pharmacotherapy. These orally delivered medications have the highest compliance, but they are plagued by severe issues such as excessive hepatic metabolism (the first-pass effect) and a lack of target specificity, resulting in a high level of side effects and toxicity. Furthermore, most medications are limited by variability in individual responses regarding weight loss, insufficient weight loss (<5% at 3 to 4 months) and cause limited sustained weight loss beyond 1 year. [3,4]. Surgery is highly effective, but it is also exceedingly intrusive and costly. Therefore, as obesity has few effective therapies, an emerging and effective therapeutic approach is urgently needed [5]. 

Browning subcutaneous white adipose tissue (WAT) using browning agents to induce beige adipocyte formation for enhanced thermogenesis is a promising approach for combating obesity [1]. WAT and brown adipose tissue (BAT) develop in different sites and have different colors, morphologies, functions, and biochemical properties [2,3]. WAT stores energy, mainly in the form of triglycerides, in a large, single (unilocular) droplet in white adipocytes. WAT is positively associated with adiposity. In contrast, brown and beige adipose tissues are metabolically active and are negatively associated with adiposity. Brown and beige adipocytes can take up fatty acids and glucose, and quickly oxidize them to produce energy as heat via non-shivering thermogenesis. This process is mediated by uncoupling protein 1 (UCP1) [4], which is exclusively expressed in the inner mitochondrial membrane of beige and brown adipocytes [5]. UCP1 dissipates the proton electrochemical gradient generated from oxidative phosphorylation in the form of heat [6]. However, the amount of brown and beige adipocytes in a human body is very limited. Luckily, WAT can be converted into beige adipocytes via browning agents. This is especially true for subcutaneous WAT, which is easier to access and is more responsive to browning agents than visceral WAT and other WAT depots. Mirabegron and CL 316,243 (selective β3 adrenergic receptor agonists), short chain fatty acid acetate, thiazolidinediones (known peroxisome proliferator-activated receptor (PPAR) agonists), exenatide and liraglutide (glucagon like peptide 1 receptor agonists), curcumin and resveratrol (polyphenols), and quercetin (flavanol) are browning agents that induce the formation and activation of beige adipocytes. Although different browning agents with potential browning effects on WAT have been suggested, their clinical application is limited due to undesirable side effects on other organs because of non-specific targeting [7,8,9]. Upregulating UCP1 is a key step in browning WAT. Mitogen-activated protein kinase, the mammalian target of rapamycin, and AMP-activated protein kinase (AMPK) signaling pathways have been reported to regulate browning [10]. AMPK is a potent signal molecule, which can activate sirtuin (silent mating type information regulation 2 homolog) 1 (SIRT1) and peroxisome proliferator-activated receptor gamma coactivator 1-alpha (PGC-1α), resulting in increased mitochondrial biogenesis and function, and upregulated UCP1 expression. 

Metformin HCl (1, 1-dimethyl biguanide hydrochloride), an antidiabetic drug, is the most potent AMPK agonist [11]. Emerging evidence has suggested that metformin might have a beneficial impact on weight loss and energy metabolism by browning WAT [12]. However, evidence is inconclusive regarding its browning and anti-obesity effectiveness, mainly due to its low bioavailability and low targeting specificity in humans [12]. The oral bioavailability of metformin is 40–60% [12]. Oral delivery is accompanied by adverse effects; in particular, gastrointestinal side effects occur in 20–30% of patients [13]. Injection of metformin to subcutaneous WAT might overcome these problems and enhance energy expenditure [14], but injection is not desirable for long-term obesity management due to poor patient compliance resulting from pain and possible infection at the injection site [15]. Thus, innovative approaches are needed to deliver metformin directly to subcutaneous WAT.

Transdermal delivery seems to be a promising alternative to the oral delivery of drugs and subcutaneous injection [16]. Microneedles (MN) and iontophoresis (INT) are among the most tested transdermal delivery approaches that are promising for drug delivery through the skin to subcutaneous WAT in a noninvasive manner [17,18]. MN arrays have been developed as a penetration enhancer, overcoming existing disadvantages related to hypodermic needle usage and improving patient compliance. Dissolving MN, commonly made from biocompatible and dissolvable polymers, are designed to dissolve in the skin and release the payload with no sharp waste left behind. Ideally, MN should be strong enough for inserting into all types of skins, and their dissolution should be controlled for sustained release of the loaded drugs from several hours to days [19,20,21]. Poly (lactic-co-glycolic acid) (PLGA) MN are strong and dissolvable and can release metformin sustainably. PLGA is an FDA-approved biocompatible and biodegradable polymer for medical devices and transdermal drug delivery systems. As a linear copolymer, PLGA can be prepared from its constituent monomers of lactic (LA) and glycolic acid (GA) at different ratios. This ratio also determines the degradation rates of PLGA, ranging from days to months in a de-esterification process. With a higher content of LA, it becomes less hydrophilic with less water absorption and subsequently degrades more slowly. The degraded monomeric components can then be removed via natural pathways. PLGA has extensive biomedical applications, including MN-based drug delivery [22,23]. 

After metformin is delivered and released into upper skin layer, it should be driven into the deeper subcutaneous WAT. Iontophoresis (INT),a popular transdermal delivery approach, utilizes an electric current, with a typical value ranging from 0.1–1.0 mA/cm^2^, as a driving force for drug permeation across the dermal layer into the subcutaneous WAT [24]. 

In the present study, we investigated the transdermal delivery of metformin directly to subcutaneous WAT using PLGA-based microneedles, assisted by ionophoresis. Their delivery efficacy and browning effectiveness in high fat diet (HFD)-induced obese C57BL/6J mice was determined. We also compared this ionophoresis-assisted microneedle (MN + INT) delivery with MN-alone and INT-alone methods. It was found that through the administration of metformin-loaded MN followed by INT (MN (met) + INT) as compared with MN (met) alone or INT (met) alone to the skins above the inguinal WAT (IgWAT), metformin could be efficiently delivered to subcutaneous WAT and further induce the browning of WAT, subsequently resulting in decreased body weight and fat gain, increased energy expenditure, and diminished fat pad size. The results suggest that INT-assisted MN transdermal delivery of metformin could be an effective approach to combating obesity and the related health problems.

## 2. Materials and Methods

The following materials were purchased: metformin and propranolol from Cayman Chemical Co. (Ann Arbor, MI, USA); PLGA, lactide:glycolide (50:50) and lactide:glycolide (75:25), and anti-UCP1, anti-AMPK, anti-pAMPK antibodies and primers from Sigma-Aldrich, Inc. (St. Louis, MO, USA); polydimethylsiloxane (PDMS) micro-molds from Micro point Technologies, Singapore; 1,1′-dioctadecyl-3,3,3′,3′-tetramethylindodicarbocyanine, 4-chlorobenzenesulfonate salt (DID), and 1,2-dioleoyl-sn-glycero-3-phosphoethanolamine-*N*-(lissamine rhodamine B sulfonyl) (ammonium salt) (Rohd-PE) dyes from Thermo Fisher Scientific Co. (San Jose, CA, USA); TRIzol reagent, Maxima First Strand cDNA Synthesis Kit, and PowerUp SYBR Green Master Mix from Thermo Fisher Scientific (Pittsburgh, PA, USA); and the Vectastain ABC kit and Vector Hematoxylin QS from Vector Laboratories (Burlingame, CA, USA). The TENS and IOMED electrodes were from Isokinetics, Inc. (De Queen, AR, USA). A Keithley 2400 current source was used as the power supply from INT Instruments Co. (Cleveland, OH, USA). 

### 2.1. MN Patch Fabrication 

MN patches were fabricated using a PDMS micro-mold casting method. The needles were arranged in a 10 × 10 array with 500 μm tip interspacing. The needles had the shape of sharp-pointed pyramids with a height of 800 μm. PLGA (50:50), dissolved in dimethylformamide (DMF) (200 mg/mL) (with or without loaded dyes or metformin) was filled into the mold using a pipette (~30–40 μL/patch). Next, filled molds were placed under a vacuum at 600 mmHg for 20 min, which allowed the solution to fill the MN cavities. Following centrifugation, a second layer of PLGA solution (75:25) (~100 μL/patch) was loaded as the backing layer in each MN mold. Loaded molds were placed in the oven at 140 °C for 30 min. MNs were cooled down at room temperature overnight. MN patches were characterized with a Hitachi 3400N VP emission scanning electron microscope (SEM). 

### 2.2. MN Patch Application Procedures

Since mice have two IgWAT depots, two MN patches and two INT patches were applied to each section of skin side above the IgWAT depots. The skin above the IgWAT was shaved and cleaned before MN application. A few drops of 3M Vetbond Tissue Adhesive were applied to the backing layer of the MN using an applicator. The MN patch was pushed into the skin by pressing with the thumb. The inserted MN patch was held (slight skin pitching) for 30 s for attachment at the place. Inserted MN were fixed and kept in situ using medical tape (surgical paper tape, 0.5”) for ~24 h. DID/metformin-loaded MN patches were inserted into mice skin using thumb pressure and left in situ for 24 h.

### 2.3. INT Application Procedures 

INT patches were cut to an appropriate size and applied to the desired skin areas. IOMEDR electrodes with reservoirs were used for applying the INT for the microneedling (MN derma-rollers) + INT and INT alone treatments, but TENS Unit Electrode pads without a reservoir was used for the MN (patch) + INT treatment. For INT alone, metformin was loaded into the patch reservoir with a pipette. In the MN + INT treatment, shortly (within a few minutes) after removing the MN patches, the adhesive electrodes were applied to the previously MN-treated skin sites (pig skin and/or shaved IgWAT area of mice). For INT, two electrode patches (cathode/anode) must be placed at the same time. The medication was loaded into cathode or anode electrode based on its carrying charges. The adhesive electrode patches then connected to the Keithley 2400 power supply. An INT current of 0.2 mA/cm^2^ was applied for up to 30 min. After that, the electrode patches and power supply were removed and disconnected.

### 2.4. In Vivo Fluorescence Imaging and Biodistribution of DID Dye Using MN Patches and INT 

Healthy 6-week-old male C57BL/6J mice (fed a chow diet) without any skin damage or health issues were used. Mice received one of the following treatments: (1) control (no MN, no INT), (2) MN (DID) alone, (3) INT (DID) alone, or (4) MN (DID) + INT. Mice received MN treatments on IgWAT skin areas (shaved) via thumb pressure. The MN and/or INT application procedures are described in the Appendix A. Mice were imaged after this treatment. Subsequently, IgWAT, liver, kidneys, GWAT, and lungs were dissected for fluorescent analysis. The IVIS^®^ Spectrum CT imaging system was used in this study to analyze all fluorescence data. 

## 3. Anti-Obesity Mice Study

### 3.1. Animals

Six-week-old male C57BL/6J mice from Jackson Laboratory were caged at 22–24 °C, 45% relative humidity, and a daily 12 h light/dark cycle, and had free access to water and diet. After 1 week of acclimation, mice were fed a HFD for 4 weeks (45% energy from fat, D12451, Research Diets, Inc., New Brunswick, NJ, USA) to induce obesity. After that, mice were weighed and randomly assigned into one of the following treatment groups (*n* = 5): 1. MN + INT (blank), 2. MN (met) + INT, 3. MN (met) alone, and 4. INT (met) alone. All treatments were given 3 times/week for an additional 5 weeks. The metformin dose was 3 mg/kg body weight/day. Metformin was dissolved in sterile saline for the INT alone treatment. MN patches and INT (at 0.2 mA/cm^2^, 10 min) were developed and applied as described above. Food intake and body weight were recorded weekly. Glucose tolerance tests (GTT) and indirect calorimetry were performed at the end of the third and fourth week of treatment, respectively. Mice were fasted overnight and euthanized humanely at the end of the fifth week of treatment. Blood was collected from the abdominal vein, and plasma was obtained by centrifugation at 1500× *g* at 4 °C for 25 min. The major organs and tissues of each mouse were collected for analysis. 

### 3.2. Body Composition

The body composition of mice was measured weekly using an EchoMRI Whole Body Composition Analyzer (EchoMRI LLC, Houston, TX, USA). 

### 3.3. Indirect Calorimetry 

Metabolic parameters were monitored (2 mice/treatment group) using a metabolic chamber system (TSE Phenomaster, TSE Systems, Inc., Chesterfield, MO, USA) for 6 days (3 days of acclimation and 3 days of data collection), including O_2_ consumption (VO_2_ (mL/h/kg)), CO_2_ production (VCO_2_ (mL/h/kg)), respiratory exchange ratio (RER), and energy expenditure (H1 (kcal/h/kg)). 

### 3.4. Plasma Lipid Profile

Plasma concentrations of total cholesterol, triglycerides, low-density lipoprotein (LDL), high-density lipoprotein (HDL), and very low-density lipoprotein (VLDL) were measured at Tufts University’s Jean Mayer USDA Human Nutrition Research Center on Aging using an AU400 clinical chemistry analyzer with enzymatic reagents (Beckman Coulter, Inc., Brea, CA, USA).

### 3.5. GTT 

For GTT, mice were fasted for 6 h and then injected intraperitoneally with glucose at a dose of 1 mg/kg body weight. Blood was taken at 0, 15, 30, 60, 90, and 120 min after injection, and blood glucose concentrations were measured with a One Touch^®^ glucometer from tail vein blood. The GTT area under the curve (GTT-AUC) was calculated by plotting blood glucose concentrations against time using the following formula:AUC=((C1+C2)/2)×(T2−T1)
where C is the glucose concentration (mg/dL) and T is the time (minutes).

### 3.6. IgWAT and Liver Metformin Content

In 1 mL of saline, IgWAT and liver tissues (about 100 mg) were homogenized. In the homogenates, 100 µL of 0.5 mg/mL propranolol-HCl (as an internal standard) and 1 mL of acetonitrile was added, followed by vortexing for 1 min. The upper aqueous phase was transferred into a fresh tube and dried under nitrogen after centrifugation at 1500× *g* for 10 min at 4 °C. The extract was reconstituted with saline and then quantified using the HPLC technique described in the Appendix A. 

### 3.7. Real-Time PCR 

TRIzol reagent was used to extract the total RNA from IgWAT and liver, and a Maxima First Strand cDNA Synthesis Kit was used to synthesis cDNA from the measured RNA. On a real-time PCR system (Eppendorf Mastercycler ep realplex instrument, Hauppauge, NY, USA), the cDNA level of the target genes was evaluated using PowerUp SYBR Green Master Mix. The 2^−ΔΔCt^ technique was used to calculate the mRNA fold changes, which were normalized against the housekeeping genes 36B4 or β-actin. 

### 3.8. Immunohistochemistry 

Immunohistochemical staining of formalin-fixed IgWAT was conducted for UCP1, AMPK, and pAMPK using the Vectastain ABC kit. The sections were then stained with hematoxylin QS counterstain. The slides were imaged using an EVOS auto-fluorescence microscope. 3,3’-Diaminobenzidine (DAB; Chromogen) on slides with hematoxylin QS was quantified according to the protocol previously developed by Nguyen et al. [25].

### 3.9. Hematoxylin and Eosin Staining

IgWAT sections were deparaffinized and rehydrated with xylene and ethanol after being embedded in paraffin (5 µm). Sections were rinsed with water to remove reagent residues, blotted to remove excess water, and then incubated with hematoxylin for 4 min before being washed with water several times. The sections were dehydrated after being stained with eosin. Finally, the portions were cleaned, and a xylene-based mounting medium was used.

### 3.10. Statistical Analysis

The statistical software packages R and SPSS_25_ were used to analyze the data. To compare the means of several groups, a one-way ANOVA was used, followed by Tukey’s HSD post hoc test. The means and standard error of the mean are used to express the data in the figures and tables. Different letters on top of the bars indicate significant differences among the various transdermal metformin treatments and the blank. Differences were considered statistically significant at *p* < 0.05.

## 4. Results

### 4.1. MN Derma-Rollers and INT-Based In Vitro Transdermal Delivery

Before metformin was incorporated into microneedles for delivery, transdermal delivery efficacy was initially determined using MN derma-rollers and INT in vitro by applying 1,2-dioleoyl-sn-glycero-3-phosphoethanolamine-*N*-(lissamine rhodamine B sulfonyl) (ammonium salt) dye to pig skin. The results (Appendix A) indicated that after perforation of the pig skin, dye molecules can be delivered to the subcutaneous WAT, driven by INT. Delivery was positively correlated with the microneedle length and the treatment time. The results indicated that ~800 μm long MN are effective for transdermal delivery.

### 4.2. The PLGA MN Patch

Figure 1A illustrates the fabrication process of dissolving PLGA MN patches using PDMS micro-molds in a mold casting method. The needles were arranged in a 10 × 10 array with 500 μm tip interspacing. They are sharp-pointed pyramids in shape, with a height of 702.7 ± 31.63 μm. The templates and the fabricated MNs are presented in Figure 1B–E (photos) and Figure 1F–I (SEM images). Metformin or DID dye was successfully loaded into the needles of PLGA MN patches. The average content of dye and metformin that was loaded into the fabricated PLGA MN patches was 90.5 ± 0.6 μg and 91.3 ± 1 μg, respectively (Appendix A; MN patch loading test). After 7 days, 98.033 ± 1.3% of loaded metformin in the MN patches was detected (Appendix A). After placing a MN patch in the dissolving medium, 40 μg (43.2%) of loaded metformin was released from the MN patch after 1 h. About 44 μg (48.4%) and 64 μg (70.1%) of the loaded metformin was released after 2 and 5 h, respectively. After 8 h, the accumulative released metformin reached 81 μg (88.7%). Dissolved MNs are shown in Appendix A (SEM images). The loaded metformin was completely released from the MN patch after 24 h (Appendix A). The delivery effectiveness of MN + INT was further evaluated in the pig skin. The permeation results of the dissolving MN patch were consistent with those from our MN derma-roller experiment. MN + INT application increased the permeation of the fluorescent dye DID through the skin layers and reached the subcutaneous WAT as compared with MN alone or INT alone (Appendix A).

### 4.3. In Vivo Fluorescence Imaging and Biodistribution of DID Dye Using MN Patches and INT

As shown in Figure 2A,B, the MN (DID) + INT treated-mice had 2- and 64-fold higher fluorescent signals in IgWAT deposits as compared with MN(DID) alone and INT (DID) alone, respectively. After dissection of the IgWAT, the MN (DID) + INT treatment had 2.5- and 2.6-fold higher cumulative fluorescent intensity in the IgWAT than MN (DID) alone or INT (DID) alone, respectively. Fluorescent signals in other organs/tissues were too low to be detected, indicating successful local delivery and less diffusion. 

#### 4.3.1. Anti-Obesity and Metabolic Benefits of Metformin in HFD-Induced Obese C57BL/6J Mice 

The anti-obesity effects of metformin delivered by MN (met) + INT were evaluated in HFD-induced obese C57BL/6J mice. There were no significant differences in food intake among all treatment groups (Figure 3A). Compared with mice in the MN + INT (blank) treatment group, mice treated with MN (met) + INT and MN (met) alone had a 3.35- and 2.2-fold decrease in body weight (*p* < 0.05). Mice treated with MN (met) + INT had a 2.9-fold decrease in body fat percentage (body fat %) compared with mice treated with MN + INT (blank) (*p* < 0.05). Changes in lean body percentage (lean body %) were 6.4- and 3.3-fold higher in mice treated with MN (met) + INT compared with mice in the MN + INT (blank) and INT (met) alone groups (*p* < 0.01) (Figure 3A). 

GTT was performed to assess whole-body glucose tolerance. While mice treated with MN (met) + INT had the lowest GTT area under the curve (GTT-AUC) and blood glucose levels among all treatment groups, the differences were not statistically significant (Figure 3B). 

Mice treated with MN (met) + INT had significantly 2.2- and 1.5-fold lower gonadal WAT (GWAT) and IgWAT mass than mice treated with MN + INT (blank) (Figure 3C). Mice treated with MN alone or INT (met) alone also had 1.4- and 1.1-fold lower GWAT and 1.4- and 1.09- fold lower IgWAT mass, but the differences did not reach statistical significance. 

Mice treated with MN (met) + INT had 1.2- and 1.3-fold higher metformin content in the IgWAT compared with mice treated with MN (met) alone and INT (met) alone (*p* < 0.001). The liver metformin content of mice treated with MN (met) + INT was 1.3-fold lower compared with mice treated with INT (met) alone (*p* < 0.05) (Figure 3D). 

Mice treated with MN (met) + INT had the lowest average adipocyte size in IgWAT among all treatment groups. The average adipocyte size in IgWAT was 3.6-, 1.5- and 2-fold lower in the MN (met) + INT group compared with mice treated with MN + INT (blank) (*p* < 0.001), MN (met) alone (*p* < 0.01), and INT (met) alone (*p* < 0.05). Furthermore, the lipid droplet morphology of mice treated with MN (met) + INT appeared more like the multilocular lipid droplet morphology of brown and beige adipocytes (Figure 3E). 

Mice treated with MN (met) + INT had the highest respiration exchange ratio (RER) among all treatment groups, reflected by the released CO_2_/consumed O_2_ ratio, which was significantly different (*p* < 0.05) from that of mice treated with MN + INT (blank) and INT (met) alone. Energy expenditure expressed as H1 (kcal/h/kg body weight), oxygen consumption expressed as VO_2_ (mL/h/kg body weight), and CO_2_ production expressed as VCO_2_ (mL/h/kg body weight) were also highest in mice treated with MN (met) + INT among the treatment groups, which were significantly different (*p* < 0.05) from those of mice treated with MN + INT (blank) and INT (met) alone (Figure 3F). 

#### 4.3.2. Browning Activities 

IgWAT UCP1 protein expression levels were 5.2- and 4.4-fold (*p* < 0.05) higher in mice treated with MN (met) + INT compared with those in mice treated with MN + INT (blank) and INT (met) alone, respectively (Figure 4A). While IgWAT AMPK protein expression levels were not different among treatment groups (Figure 4B), IgWAT phosphorylated AMPK (pAMPK) protein expression levels were 13.3- and 6.3-fold (*p* < 0.05) higher in mice treated with MN (met) + INT compared with those in mice treated with MN + INT (blank) and INT (met) alone, respectively (Figure 4C).

IgWAT mRNA levels of brown/beige fat-specific markers (UCP1, 11.9-fold), (ELOVL3, 17-fold), (PRDM16, 8.7-fold), (TMEM26, 2.6-fold), (CIDEA, 12-fold), (ZIC1, 5-fold), (PGC1α, 3.5-fold), and (CD137, 2.3-fold) (*p* < 0.001) were higher in mice treated with MN (met) + INT compared with those in mice treated with MN + INT (blank). Furthermore, IgWAT UCP1, ELOVL3, PRDM16, CIDEA, PGC1α, and CD137 mRNA levels were also significantly higher (*p* < 0.05) than in mice treated with MN (met) or INT (met) alone (Figure 4D). 

The mRNA expression levels of the IgWAT lipogenic genes acetyl-CoA carboxylase 1 (ACC1) (7-fold) and leptin (4.5-fold) (*p* < 0.05) were lower in mice treated with MN (met) + INT compared with those in mice treated with MN + INT (blank) (Figure 4E). Similarly, the mRNA expression levels of the IgWAT inflammatory markers MCP1 and TNFα were 6.9- (*p* < 0.001) and 6.6-fold (*p* < 0.01) lower in mice treated with MN (met) + INT compared with the MN + INT (blank) group (Figure 4E). The mRNA expression levels of the liver inflammatory markers MCP1 and TNFα were 3.9-fold (*p* < 0.01) and 11.7-fold (*p* < 0.001) lower in mice treated with MN (met) + INT compared with mice treated with MN + INT (blank) (Figure 4F). The mRNA expression levels of phosphoenolpyruvate carboxykinase (PEPCK) and glucose transporter type 2 (GLUT2) in MN (met) + INT mice were also 2.3-fold (*p* < 0.01) and 4.7-fold (*p* < 0.001) lower compared with those in mice treated with MN + INT (blank) (Figure 4F). 

There were no significant differences in the plasma lipid profiles, including total cholesterol, triglycerides, LDL, HDL, and VLDL among all treatment groups.

## 5. Discussion

As the most common delivery routes for anti-obesity therapeutics, oral administration and injections are limited by the low oral bioavailability, high hepatic metabolism, adverse side effects, and high responsive doses of therapeutics, along with high levels of variability, discomfort, and pain at injection sites [26,27]. Since subcutaneous WAT is easy to access and has a higher browning potential than other WAT depots, transdermal delivery locally to subcutaneous has emerged as an attractive alternative for obesity treatment [14,28]. MN-assisted delivery of browning agents to subcutaneous WAT has been demonstrated successfully in animal studies [14,28]. The effective dose of browning agents when administered transdermally via hyaluronic acid-based MN patches to subcutaneous WAT was much lower compared with that of conventional oral or systemic injection. Delivery of thyroid hormone T3 at 0.5 g/day for 5 days via a MN patch had the same or better anti-obesity effect compared with ~12.5 g/day systemic injection for 2 weeks in mice [14]. MN-assisted IgWAT delivery of CL 316,243 at 1 mg/day to IgWAT for 4 weeks [28] or 5 mg/day for 5 days via a MN patch [14] induced the browning of subcutaneous WAT and inhibited body weight gain by ~15% compared with the control mice treated with drug-free MN patches. However, intraperitoneal injection of the same dose did not significantly prevent weight gain [14]. These studies demonstrated that MN patches were more efficacious than conventional routes for delivering drugs to subcutaneous WAT, further promoting WAT browning, resulting in body weight and fat loss as well as improvements in metabolic health, including glucose homeostasis and the lipid profile [28,29]. While few studies have highlighted the application of dissolving MN for transdermal delivery of browning agents [14,28], to date, there has been no investigation of the influence of the combination of MN and INT for subcutaneous WAT delivery of these agents. INT improves the transdermal delivery of polar hydrophilic therapeutic molecules by approximately 10–2000 times more compared with their conventional application to the skin surface [30]. Indeed, it has been suggested that the combination of MN and INT leads to an increase in the number of therapeutic molecules delivered by dissolving MN patches, and thus their delivery efficacy is higher compared with each of them alone [15,31]. As such, for the first time, this study reports on the synergistic effects of MN + INT application to enhance the WAT browning efficiency and metabolic health benefits of metformin. 

Metformin is an FDA-approved, low-potency, high-dose drug for Type 2 diabetes [32]. When orally administered at doses of 500–3000 mg/day, blood metformin concentrations are less than 40 μM in human subjects, which cannot induce the browning of WAT. However, overdoses with metformin can have severe complications [26,27]. The oral route with a relatively high therapeutic dose of 200 mg/kg/day [33] or 250 mg/kg/day [34], a dose that is approximately seven times higher the maximal daily dose for human [35], was used in previous studies in mice that examined the effect of metformin on metabolism and the browning of adipose tissue. The 3 mg/kg body weight/day metformin dose in this study is more than 60 times lower than the previously reported oral therapeutic dose of metformin in obese mice, and is equivalent to an oral dose of ~205–220 mg/day in humans (~12 times lower the maximum therapeutic dose of metformin in a man weighing 70 kg, i.e., approximately 35 mg/kg/day) [36,37]. More prominent dose reductions are anticipated in humans, as they have more subcutaneous WAT (~10% of body weight) than mice (<1% of mouse IgWAT) [14]. In addition to the dose advantages of transdermal delivery of metformin, the MN patch also gave sustained delivery of metformin [38]. Our PLGA MN patch consistently released and delivered metformin for 24 h. 

Our results showed that when metformin was delivered to IgWAT via MN + INT, mice had decreased body weight and visceral fat, enhanced browning of WAT, increased energy expenditure, improved glucose homeostasis, and suppressed obesity-induced inflammation. Consistently, IgWAT weight was reduced in the mice treated with MN (met) + INT. In addition, GWAT weight was also significantly reduced, which can possibly be attributed to body weight loss [28]. MN (met) + INT treatment increased RER and oxygen consumption, and enhanced energy expenditure as an outcome of induced IgWAT browning. The high RER indicated an increase in fatty acid utilization that might be due to the browning of WAT [28]. Fatty acid oxidation provides the mitochondrial bioenergetics as well as being the biophysical activator of UCP1-induced uncoupling [39]. The IgWAT of mice treated with MN (met) + INT had smaller adipocytes with a more multilocular appearance. These changes were accompanied by increased gene expression of UCP1, ELOVL3, PRDM16, TMEM26, CIDEA, ZIC1, PGC-1α, and CD137 in IgWAT. These data supported the idea that MN + INT transdermal delivery of metformin induced IgWAT browning. 

Metformin accumulation in WAT facilitates insulin-medicated glucose uptake, utilization, and metabolism. On the other hand, it has been shown that metformin induces the expression of BAT and beige adipose tissue-associated genes such as UCP1 and PRDM16, and enhances mitochondrial biogenesis, thermogenesis, and fatty acid uptake [33]. While there has been no previous study comparing the metformin content in the liver and WAT following transdermal delivery, in this study, although metformin content in the liver was lower in mice treated with MN (met) + INT compared with the other treatment groups, they showed lower expression levels of liver inflammatory, lipogenic and glucogenic markers. The local dominant accumulation of metformin in IgWAT of mice treated with MN (met) + INT was further supported the IgWAT browning effect of metformin when delivered locally via MN + INT. 

Dysregulation and increased lipolysis of subcutaneous WAT in HFD-induced C57BL/6J obese mice have been shown previously [40]. The availability of free fatty acids and glycerol is also associated with the stimulation of hepatic gluconeogenesis and increased blood glucose levels [41]. Lipolysis-derived fatty acids released from lipid droplets act as fuel for non-shivering thermogenesis and UCP1-mediated thermogenesis and heat production [42]. Indeed, it is well known that under activated thermogenesis, the ability for the uptake and utilization of glucose and lipids by brown/beige adipocytes is increased, which further explain the promising metabolic benefits of WAT browning in the regulation of the glucose metabolism [41]. MN (met) + INT treatment improved glucose homeostasis more efficiently compared with other treatment groups during GTT. Improvements in glucose tolerance and downregulation of glucogenic genes including GLUT2 and PEPCK in the liver, a primary site of action of metformin as an inhibitor of hepatic gluconeogenesis [35], further indicate the enhanced effects of MN (met) + INT treatment on improvements in glucose metabolism, which are possibly caused by browning rather than just the anti-hyperglycemic and insulin-sensitizing effects of metformin. 

The gene expression levels of ACC1 and leptin in IgWAT were downregulated in the MN (met) + INT treatment group compared with the other treatment groups. Generally, WAT-released free fatty acids (produced from adipocyte lipolysis) are taken up by the liver, which is then involved in the production of triglyceride-rich lipoproteins, VLDL, and downstream lipoprotein metabolism. On the other hand, the liver is strongly associated with the regulation of adipose tissue metabolism for delivering triglycerides to adipocytes. Thermogenic brown/beige adipocytes use it for adaptive thermogenesis and heat production. Thermogenic adipocytes continuously replenish these sources of fuels by internalizing triglyceride-derived fatty acids provided by the liver; furthermore, fatty acids also become available through cellular uptake, de novo lipogenesis, and from multilocular lipid droplets in brown adipocytes [43,44]. Decreased total cholesterol, triglycerides, and LDL plasma levels have been suggested as indicators of improvements in obesity-related metabolic disorders [45], but their plasma levels did not reach statistically significant differences in this study. While there is no study that has specifically determined the effect of local delivery of metformin to subcutaneous WAT on lipid and lipoprotein metabolism, previous studies where metformin was delivered orally indicated that metformin’s effects on lipid metabolism seem to be localized to the intestine and liver, where metformin could affect lipoprotein metabolism, resulting in decreased plasma total cholesterol, triglyceride, and LDL levels [46]. 

Mice treated with MN (met) + INT had increased pAMPK protein levels in IgWAT. Metformin-mediated activation of AMPK is linked to the regulation of several of the key proteins involved in lipid and glucose metabolism [47]. ACC1 is one of the major downstream targets of AMPK that becomes phosphorylated and inactivated, and results in decreased fatty acid synthesis and increased fatty acid β-oxidation. This itself improves adipose tissue metabolism because of decreased ectopic fat accumulation [48]. In addition to ACC1, GLUT2 and PEPCK are also important metabolic targets of AMPK in adipose tissue and the liver [49]. Metformin induced the activation of AMPK, which suppressed the transcription of the liver enzymes GLUT2 and PEPCK, which resulted in the inhibition of gluconeogenesis [50]. Furthermore, evidence has shown that metformin-activated AMPK is associated with enhanced expression levels of UCP1 and may improve brown adipogenesis and thermogenesis [51,52].

Since obesity and the related comorbidities are associated with chronic low-grade inflammation [53], the expression levels of proinflammatory markers in IgWAT were also examined. MN (met) + INT treatment suppressed the expression of MCP1 and TNFα in IgWAT. Brown and white adipocytes play important roles in the modulation of the inflammatory response in a way that increases in the brown adipocytes in adipose organs can suppress the proinflammatory phenotype, characterizing the development of healthier tissue, consequently improving energy metabolism, and reducing obesity [54]. These improvements are not only because of phenotypic changes in the adipose tissue (browning of WAT, more brown adipocytes) but also simple subcutaneous WAT size reductions could improve inflammation [55]. Subcutaneous WAT influences the inflammatory and metabolic status in the liver through regulating the production and release of free fatty acids, lipid mediators, and inflammatory cytokines. Current evidence suggests that brown and beige adipocytes also improve the liver’s inflammatory response [43,56]. Liver expression levels of MCP1 and TNFα decreased in the MN (met) + INT treatment group, which further indicated that the browning of subcutaneous WAT decreased not only WAT but also liver inflammation. 

## 6. Conclusions

Taking all the findings together, our study indicated that MN + INT effectively delivered metformin to subcutaneous WAT and further induced the browning of subcutaneous WAT, resulting in body weight and fat loss, and enhanced metabolic health. This MN + INT transdermal delivery approach has a potent dose advantage compared with traditional oral or intravenous delivery approaches, and thus may have potential to manage obesity in an effective, easy, and safe regimen.

## Figures and Tables

**Figure 1 pharmaceutics-14-00879-f001:**
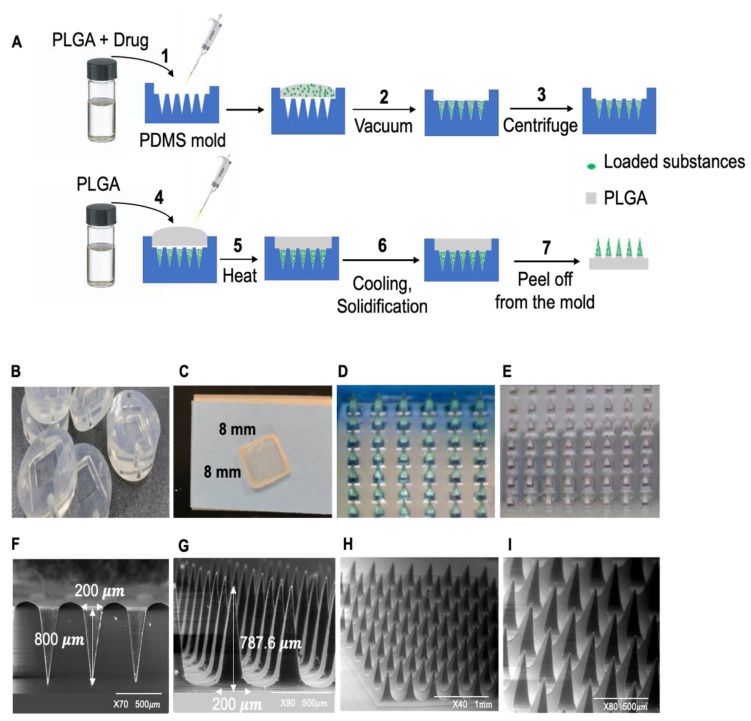
Microneedle fabrication and characteristics. (**A**) Schematic illustration of the process of fabricating dissolving PLGA MN patches. (**B**) MPatch microneedle mold. (**C**) Fabricated MN patch overview. (**D**) DID-loaded MN and (**E**) Rhod-PE-loaded MN. SEM micrographs of MN patches. (**F**) Length and base of the mold. (**G**) Length and base of the needles in 800 μm MN. (**H**) Intact 800 μm MN. (**I**) Intact 800 μm MN.

**Figure 2 pharmaceutics-14-00879-f002:**
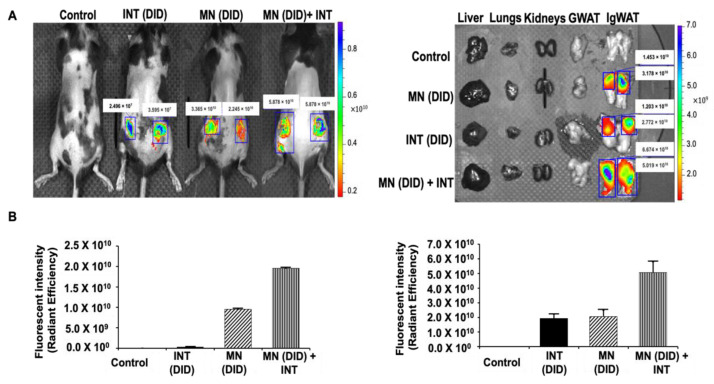
In vivo fluorescence imaging and biodistribution of DID dye using MN patches and INT. IVIS images of (**A**) the biodistribution of DID dye using different treatments: Control, INT (DID) alone, MN (DID) alone, and MN (DID)+ INT in C57BL/6J mice and dissected organs. (**B**) Quantified fluorescence intensity of the delivered DID dye. Images are representatives of three independent experiments. Data were calculated from three independent experiments and expressed as means ± SEM.

**Figure 3 pharmaceutics-14-00879-f003:**
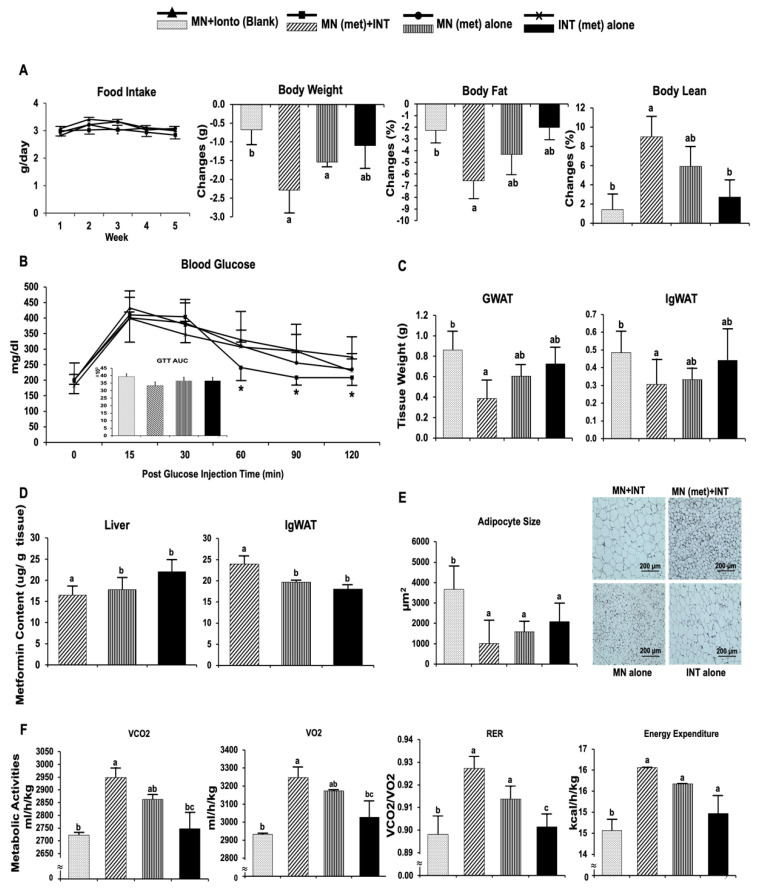
Anti-obesity and metabolic effects of metformin. (**A**) Food intake, body weight, fat %, and lean %; (**B**) blood glucose levels and glucose tolerance test area under the curve; (**C**) tissue weight of GWAT and IgWAT; (**D**) metformin content in the liver and IgWAT; (**E**) adipocyte size; (**F**) metabolic activity parameters. (*) Lower blood glucose. Values are means ± SEM, *n* = 5 per treatment. Bars without a common superscript differ at *p* < 0.05.

**Figure 4 pharmaceutics-14-00879-f004:**
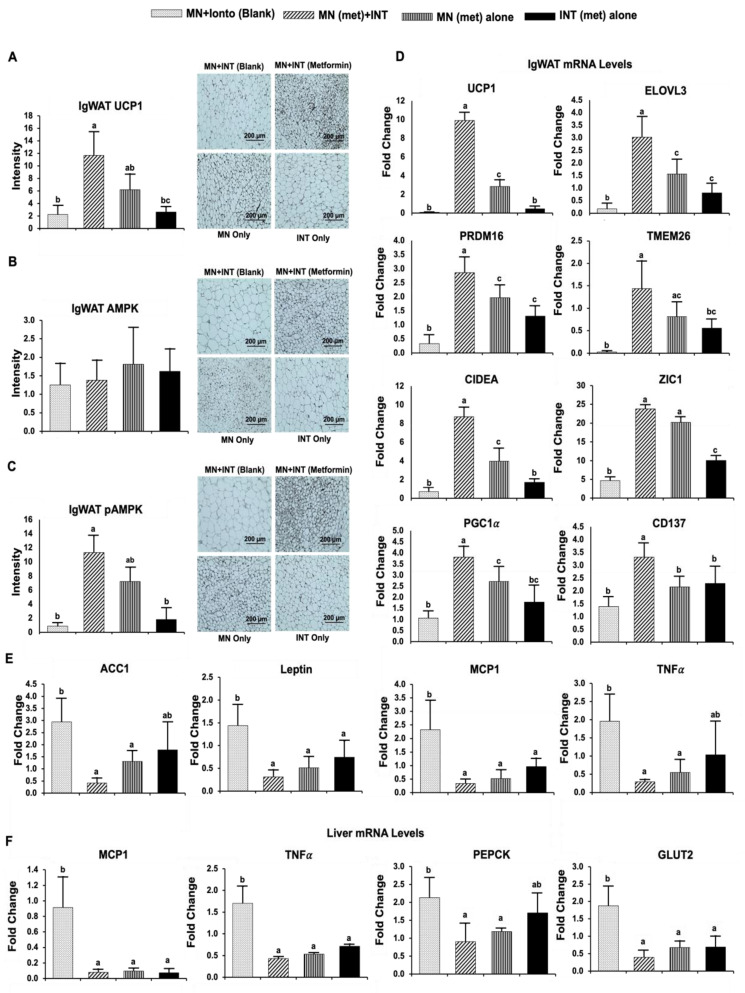
Browning effects, and thermogenic, lipogenic, inflammatory and glucogenic gene expression. Reciprocal intensity of the chromogen staining and representative immunohistochemistry staining images of IgWAT (scale bar: 200 µm): (**A**) UCP1, (**B**) AMPK, and (**C**) pAMPK. IgWAT mRNA level of (**D**) thermogenic markers: UCP1, ELOVL3, PRDM16, TMEM26, CIDEA, ZIC1, PGC1α, and CD137; (**E**) lipogenic (ACC1 and leptin) and inflammatory (MCP1 and TNFα) markers. Liver mRNA levels of (**F**) inflammatory (MCP1 and TNFα) and glucogenic (PEPCK and GLUT2) markers. Values are means ± SEM, *n* = 5 per treatment. Bars without a common superscript differ at *p* < 0.05.

## Data Availability

Not applicable.

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
