# Peer review of "Transdermal Delivery of Metformin Using Dissolving Microneedles and Iontophoresis Patches for Browning Subcutaneous Adipose Tissue"

_pharmaceutics, 2022, doi:10.3390/pharmaceutics14040879_

Round 1

Reviewer 1 Report

In the proposed paper, titled “Transdermal delivery of metformin using dissolving microneedle and iontophoresis patches for browning subcutaneous adipose tissue”, the authors presented in vitro and in vivo studies in which they tested a transdermal delivery system, composed by a dissolvable microneedles patch, coupled with iontophoresis, in order to deliver metformin to the subcutaneous white adipose tissue in obese mice. The authors investigated the browning of white adipose tissue induced by metformin and its subsequent thermogenesis effects. Moreover, they compared the browning effect induced by their delivery method (microneedles patch and iontophoresis) to microneedles alone and iontophoresis alone; they founded that their methods has better antiobesity activity in term of decreasing body weight and fat gain, increasing energy expenditure, decreasing fat pad size, and improving energy metabolism through browning of white adipose tissue.

In this manuscript, the approach proposed is explained in detail, the experiments are reproduced very well and yield statistically firm results. Moreover, the in vivo experiments are the real strengths of this paper.

The manuscript might be very interesting and relevant to the pharmaceutical, biotechnological, and, more in general, for the whole medical community, however, it requires several modification before publication on Pharmaceutics. There are some relevant formal and technical issues that must be addressed.

  1. The article is mainly well written, anyway, the authors should check punctuation, especially in the introduction sections and in materials and methods.

  1. The authors should revise the entire introduction to make the speech more homogeneous and, then, understandable for the readers. The subsections are in fact unrelated to each other. The authors could add a few sentences to connect the various sections, in order to make the introduction more understandable.

  1. In the lines 56 -57, the authors mentioned browning agents. Can you report more details about them?

  1. The article is full of abbreviations and acronyms. Usually, the use of the acronyms should facilitate the reader but, not in this case. A list of abbreviations must be inserted at the beginning of the manuscript.
  2. What are the height of the microneedles? The authors indicate a height of about 800 microns but should better characterize the needles and report a more accurate measurement, expressed as mean value and standard deviation.

  1. The color red / green / blue in the scheme of the figures reported both in the manuscript and in the supplementary materials is unreadable for color-blind people. This is no longer acceptable in today’s publishing. The authors may want to consider the popular scientific coloring guides in line.

  1. All the figures in the manuscript must be revised in depth. More in details, the authors should increase the size of the letters and numbersand the error bars are hidden behind the bars in the histogram; in the Figure 2, the letters A and B are missing.

Author Response

March 23, 2022 

Dear Editors: 

I am writing to re-submit the manuscript entitled “Transdermal delivery of metformin using dissolving microneedle and iontophoresis patches for browning subcutaneous adipose tissue” to Pharmaceutics. The original manuscript ID is pharmaceutics-1644551. I described the all changes in the revised manuscript at the end of this letter and responded to reviewer comments in the following document entitled “Response to Reviewers’ Comments”. 

Response to reviewers' comments: 

(The manuscript ID: pharmaceutics-1644551) 

We are very much thankful to the reviewers for their deep and thorough review. We have revised our present research manuscript in the light of their useful suggestions and comments. We hope that this revision has improved the manuscript to a satisfactory level. 

 Reviewer #1: 

In this manuscript, the approach proposed is explained in detail, the experiments are reproduced very well and yield statistically firm results. Moreover, the in vivo experiments are the real strengths of this paper. The manuscript might be very interesting and relevant to the pharmaceutical, biotechnological, and more in general, for the whole medical community, however, it requires several modifications before publication on Pharmaceutics. There are some relevant formal and technical issues that must be addressed. 

1. The article is mainly well written, anyway, the authors should check punctuation, especially in the introduction sections and in materials and methods. 

Response: The punctuations are checked and revised accordingly. Please see revised manuscript. 

2. The authors should revise the entire introduction to make the speech more homogeneous and then, understandable for the readers. The subsections are in fact unrelated to each other. The authors could add a few sentences to connect the various sections, in order to make the introduction more understandable. 

Response: The introduction section is revised accordingly. Please see revised manuscript. 

3. In the lines 56 -57, the authors mentioned browning agents. Can you report more details about them? 

Response: A paragraph describing the browning agents is added. Please see revised manuscript. 

4. The article is full of abbreviations and acronyms. Usually, the use of the acronyms should facilitate the reader but, not in this case. A list of abbreviations must be inserted at the beginning of the manuscript. 

Response: A list of abbreviations is inserted at the beginning of the manuscript. Please see revised manuscript. 

5. What are the height of the microneedles? The authors indicate a height of about 800 microns but should better characterize the needles and report a more accurate measurement, expressed as mean value and standard deviation. 

Response: Based on our SEM images the height of the microneedle is 702.7 ± 31.63 ?m. Please see the section of "3.2. The PLGA MN patch" in the revised manuscript. 

6. The color red / green / blue in the scheme of the figures reported both in the manuscript and in the supplementary materials is unreadable for color-blind people. This is no longer acceptable in today’s publishing. The authors may want to consider the popular scientific coloring guides in line. 

Response: Figures are revised. Please see revised manuscript for the corrected images. 

7. All the figures in the manuscript must be revised in depth. More in details, the authors should increase the size of the letters and numbers and the error bars are hidden behind the bars in the histogram; in the Figure 2, the letters A and B are missing. 

Response: We increased all sizes by 2 (e.g., charts titles from 20 to 22, numbers and letters 24 to 28 and 18 to 20, respectively). The width of the error bars is increased from 2 to 2.5 pt. The letters A and B in the figure 2 are added. Please see revised manuscript for the corrected images. 

Reviewer 2 Report

In section 3.1 appears results from derma rollers. Is not clear why is presented the data from MN derma rollers? Are the needles produced by PLGA? This need to be clarified.

I would suggest you to move from supplementary file to main text the figure S1.

Line 259 which was the dissolving medium? Add it to here.

The Figure S2C Metformin-loaded MN patch release profile, which release kinetic can be adjusted to your release assay and which is the mechanism associated that controls the release. Discuss this data.

A permeation assay ex-vivo using the pork skin would give valuable data how drug is released from needles.

Author Response

March 23, 2022 

Dear Editors: 

I am writing to re-submit the manuscript entitled “Transdermal delivery of metformin using dissolving microneedle and iontophoresis patches for browning subcutaneous adipose tissue” to Pharmaceutics. The original manuscript ID is pharmaceutics-1644551. I described the all changes in the revised manuscript at the end of this letter and responded to reviewer comments in the following document entitled “Response to Reviewers’ Comments”. 

Response to reviewers' comments: 

(The manuscript ID: pharmaceutics-1644551) 

We are very much thankful to the reviewers for their deep and thorough review. We have revised our present research manuscript in the light of their useful suggestions and comments. We hope that this revision has improved the manuscript to a satisfactory level. 

Reviewer #2: 

The authors presented a detailed data-oriented manuscript on the possibility of combining MN with INT for a more effective delivery media for anti-obesity applications. I have few comments to make. 

1. Line 243. correct the parenthesis use. 

Response: Corrected as 2-dioleoyl-sn-glycero-3-phosphoethanolamine-N- (lissamine rhodamine B sulfonyl) (Rhod-PE). 

2. In figure 3, what are the label a, b, and ab refer to? 

Response: Different letters on top of the bars indicate significant differences among various forms of metformin transdermal treatments and blank. For instance, "a" means MN (met)+INT group has significant different from treatment groups labeled as "b", the group labeled as "ab" indicates that this group doesn’t have any significant different with any of groups labeled as "a" or "b". 

3. What are the percentage reductions in the body weight of the treated mice? 

Response: The percentage reductions in the body weight of the treated mice are: 

MN + INT (blank) = -2.32 

MN (met) + INT = -8.23 

MN (met) alone = -5.35 

INT (met) alone = -3.88 

Reviewer 3 Report

The authors presented a detailed data oriented manuscript on the possibility of combining MN with INT for a more effective delivery media for anti-obesity applications. I have few comments to make.

Line 243. correct the parenthesis use.

In figure 3, what are the label a, b, and ab refer to?

What are the percentage reductions in the body weight of the treated mice?  

Author Response

March 23, 2022 

Dear Editors: 

I am writing to re-submit the manuscript entitled “Transdermal delivery of metformin using dissolving microneedle and iontophoresis patches for browning subcutaneous adipose tissue” to Pharmaceutics. The original manuscript ID is pharmaceutics-1644551. I described the all changes in the revised manuscript at the end of this letter and responded to reviewer comments in the following document entitled “Response to Reviewers’ Comments”. 

Response to reviewers' comments: 

(The manuscript ID: pharmaceutics-1644551) 

We are very much thankful to the reviewers for their deep and thorough review. We have revised our present research manuscript in the light of their useful suggestions and comments. We hope that this revision has improved the manuscript to a satisfactory level. 

Reviewer #3: 

The submitted work deals with an interesting topic concerning the improvement of the effectiveness of metformin therapy in the case of health problems and obesity. In mice studies, the authors tested the transdermal delivery of metformin directly to the subcutaneous WAT by means of PLGA-based microneedles, assisted by iontophoresis. The results obtained suggest that the transdermal delivery of metformin assisted by iontophoresis may be an effective approach to combating obesity and improving health. The presented work is interesting and valuable and should be continued towards the applicability of the described methodology in humans. 

Reviewer 4 Report

The submitted work deals with an interesting topic concerning the improvement of the effectiveness of metformin therapy in the case of health problems and obesity. In mice studies, the authors tested the transdermal delivery of metformin directly to the subcutaneous WAT by means of PLGA-based microneedles, assisted by iontophoresis. The results obtained suggest that the transdermal delivery of metformin assisted by iontophoresis may be an effective approach to combating obesity and improving health. The presented work is interesting and valuable, and should be continued towards the applicability of the described methodology in humans.

Author Response

March 23, 2022 

Dear Editors: 

I am writing to re-submit the manuscript entitled “Transdermal delivery of metformin using dissolving microneedle and iontophoresis patches for browning subcutaneous adipose tissue” to Pharmaceutics. The original manuscript ID is pharmaceutics-1644551. I described the all changes in the revised manuscript at the end of this letter and responded to reviewer comments in the following document entitled “Response to Reviewers’ Comments”. 

Response to reviewers' comments: 

(The manuscript ID: pharmaceutics-1644551) 

We are very much thankful to the reviewers for their deep and thorough review. We have revised our present research manuscript in the light of their useful suggestions and comments. We hope that this revision has improved the manuscript to a satisfactory level. 

Reviewer #4: 

1. In section 3.1 appears results from derma rollers. Is not clear why is presented the data from MN derma rollers? Are the needles produced by PLGA? This need to be clarified. 

Response: 

We hypothesized that MN + INT could further improve the delivery of dye/metformin to subcutaneous WAT. To test if such combined application will enhance the permeation through the skin layers, we did an in vitro test. In vitro permeation study was performed using pig abdominal skin. MN derma roller was used for microneedling. INT was applied using iontophoretic patches and electrodes connected to a constant-current power supply. We demonstrated that MN + INT application enhanced the permeation across the skin layers from epidermis to subcutaneous WAT in vitro and confirmed the delivery effectiveness and increased of permeation with time and MN length. After initial hypothesis testing (in vitro) using MN derma rollers, we have successfully developed, fabricated, and tested dissolving MN patches. 

This has been described in the result section 3.1 as: 

"3.1. MN derma rollers and INT based in vitro transdermal delivery 

Before metformin was incorporated into microneedles for delivery, transdermal delivery efficacy was initially determined using MN derma rollers and INT in vitro, by applying 2-dioleoyl-sn-glycero-3-phosphoethanolamine-N- (lissamine rhodamine B sulfonyl) (Rhod-PE) dye on the pig skin. The results (Figure S1) indicated that after perforation of pig skin, dye molecules can be delivered to the subcutaneous WAT under INT driving. The delivery is positively correlated with the microneedle length and the treatment time. The result indicates that ~ 800 μm long MN is effective for transdermal delivery." 

2. I would suggest you move from supplementary file to main text the figure S1. 

Response: Initially, we used MN derma rollers to evaluate whether the length of MN could improve the delivery of dye to subcutaneous WAT and to evaluate if combined MN + INT application could enhance the permeation through the skin layers. It might make readers confused if we move the data of MN derma rollers to the main text. To make manuscript simpler and easier to follow, we put the initial MN derma rollers and INT testing in the supplementary materials. 

3. Line 259 which was the dissolving medium? Add it to here. 

Response: The medium contained 1XPBS: ethanol (10:1, pH 7.4, 37°C). Please see supplementary materials > MN patch loading test > Recovery of loaded dye/drug released into the medium. 

4. The Figure S2C Metformin-loaded MN patch release profile, which release kinetic can be adjusted to your release assay and which is the mechanism associated that controls the release. Discuss this data. 

Response: 

The dissolving MN patch is made of two different types of poly (lactic-co-glycolic acid) (PLGA); lactide: glycolide (50:50), and lactide: glycolide (75:25). PLGA is a biocompatible and biodegradable polymer for medical devices and transdermal drug delivery systems. As a linear copolymer, PLGA can be prepared with its constituent monomers of lactic (LA) and glycolic acid (GA) at different ratios. This ratio also determines the degradation rates of PLGA ranging from days to months in a de-esterification process. With a higher content of LA, it becomes less hydrophilic with less water absorption, and subsequently degrades more slowly. The degraded monomeric components can then be removed via natural pathways. PLGA has extensive biomedical applications, including MN-based drug delivery. We used 50:50 for tips to fasten the dissolution and 75:25 as backing layer. Drug is release from PLGA MN patch due to polymer erosion and release drug from polymer matrix and diffusion through the skin layers as we could see the dissolved MN tips after application via SEM images (Figure S2, A and B). 

5. A permeation assay ex-vivo using the pork skin would give valuable data how drug is released from needles. 

Response: 

Please see supplementary materials > 2. Recovery of loaded DID/metformin delivered to the skin upon PLGA MN application: 2.1 and 2.2., and Release profile of the metformin-loaded PLGA MN patch in vitro. 

In addition, we have also conducted below study previously, however data are not presented in the manuscript as it was part of the initial testing: 

"In vitro skin permeation study of metformin: 

To test the effect of MN and iontophoresis in the delivery of metformin across the skin layers we did following metformin permeation study: 

Method: Permeation of metformin across pig skin was examined by comparing the following treatments: 1. Metformin solution topically applied over the skin (no MN or iontophoresis), 2. Metformin-loaded MN alone, and 3. Metformin-loaded MN in combination with iontophoresis. For treatment 1 (as control), metformin solution volume was adjusted to be consistent with the amount of drug loaded into the MN, while for the second treatment (MN only), metformin MN was applied over the skin (MN left in situ for 24 h, and incubation at 37℃). For the third treatment (MN with iontophoresis) after PLGA MN application (MN left in situ for 24 h, and incubation at 37℃) and removal, iontophoresis (0.2 mA/cm2, 30 minutes) was applied on the MN treated area with iontophoretic electrodes connected to power supply. After treatments, delivered metformin was quantified using HPLC. 

Results: Results from metformin permeation study were consistent with our results from dye permeation across pig skin using MN derma rollers, and MN and iontophoresis in vitro studies. Following combined MN and iontophoresis application; the amount of metformin recovered from the pig skin was 4.8% and 3.5% higher compared to treatment 1 and 2, respectively". 

# About duplication: 

Response: 

1. Highlighted paragraphs and sentences were paraphrased accordingly in the revised manuscript. 

2. Some of the highlighted phrases are common name of some materials as we are not able to paraphrase them as: 

Name of dyes, kits, and reagents: 

• 1,1′-dioctadecyl-3,3,3′,3′- tetramethylindodicarbocyanine, 4-chlorobenzenesulfonate salt (DID) 

• 1,2-dioleoyl-sn-glycero-3-phosphoethanolamine-N- (lissamine rhodamine B sulfonyl) (ammonium salt) (Rohd-PE) dyes. 

• Maxima First Strand cDNA Synthesis Kit. 

• TRIzol® reagent. 

• PowerUp SYBR™ Green Master Mix from Thermo Fisher Scientific (Pittsburgh, PA, USA). 

• Vectastain ABC kit and Vector Hematoxylin QS from Vector Laboratories. 

Round 2

Reviewer 1 Report

I would like to thank you the Author for the point-by-point answers. I really appreciated all the modification in the manuscript. 

Unfortunately, in the modified version of the paper it is impossible to correctly see the graphs and the images because the new images are superimposed on the old ones. for this reason it is impossible to correctly observe figures.

 Then, the authors should provide a new version of the manuscript in which the images are clearly visible.